# Early cellular events of osteomucosal healing in the tooth extraction socket

Sol Kim[1,2], Minju Song[1,2,3], Drake Williams[1,2], Wen Du[1,4], Inwoo Cho[1,5], Eunbin Bae[1,2], Joey Kim[1], Ahana Goswami[1], Ki-Hyuk Shin[1,6], No-Hee Park[1,6,7], Reuben H. Kim[1,2,6]*

1 The Shapiro Family Laboratory of Viral Oncology and Aging Research, UCLA School of Dentistry, Los Angeles, California, United States of America, 2 Section of Restorative Dentistry, UCLA School of Dentistry, Los Angeles, California, United States of America, 3 Department of Conservative Dentistry, National Health Insurance Service Ilsan Hospital, Goyang, Republic of Korea, 4 State Key Laboratory of Oral Diseases, National Clinical Research Center for Oral Diseases, Department of Prosthodontics, West China Hospital of Stomatology, Sichuan University, Chengdu, P.R. China, 5 Department of Periodontology, School of Dentistry, Dankook University, Cheonan, Korea, 6 UCLA Jonsson Comprehensive Cancer Center, Los Angeles, California, United States of America, 7 Department of Medicine, David Geffen School of Medicine at UCLA, Los Angeles, California, United States of America

* rkim@dentistry.ucla.edu

## Abstract

Healing after dentoalveolar trauma, such as tooth extraction, is unique to the oral cavity that involves osteomucosal healing – healing of soft and hard tissues at the same time – through a series of healing stages. The healing process of soft or hard tissues is well-documented previously; however, inter-dependency and cross-talks during the progression of their simultaneous healing processes remain unclear. In this study, we investigated spatial and temporal changes of epithelial, connective, and bone tissues, as well as the presence of osteoclasts, during the early stages of osteomucosal healing. We extracted the maxillary first molars in mice and examined the osteomucosal healing process daily for 7 days using histology, immunohisto-chemistry, and micro-computed tomography (microCT). Epithelial tissues closed progressively throughout 7 days. Collagen deposition began in the extraction sockets as early as day 2, forming a scaffold essential for both epithelial tissue closure and bone formation. Osteoclasts appeared on day 2, steadily increasing until day 5 and remained around the socket walls but not within the sockets. Woven bone formed rapidly around day 5, with significant mineralization observed by day 7. Notably, we identified elevated expression of RANKL throughout the process and a sharp increase in OPG near new bone on day 6. These findings demonstrated the sequential and coordinated mechanisms underlying early osteomucosal healing and provide novel insights into the critical early steps required for proper healing in the oral cavity.

**Data availability statement:** All relevant data are within the paper and its Supporting Information files.

**Funding:** This study was supported by grants from the National Institute of Dental and Craniofacial Research (R01DE023348 and R56DE033281 to Dr. R.H.K.).

**Competing interests:** The authors have declared that no competing interests exist.

## Introduction

The wound healing process involves a series of distinct yet interconnected stages [1]. In the soft tissues such as skin, wound healing after lacerations occurs through the four distinct stages, including hemostasis, inflammation, proliferation, and remodeling [2–4]. Immediately after an injury, key elements of the coagulation cascade are activated, genes associated with inflammation are stimulated, and specialized immune cells are summoned to halt bleeding and thwart infection. Upon the formation of platelets and the subsequent cessation of bleeding, a fibrin matrix is established. This matrix serves as a scaffold for the proliferation and development of new tissue and facilitates wound healing. Finally, tissues undergo remodeling steps under which granulation tissue matures into scar by increasing tensile strength and diminishing cellular contents.

On the other hand, the wound healing process of the hard tissues, such as bone, occurs in similar patterns but with differently defined stages that include hematoma, inflammation, callus formation, and remodeling [5–7]. One of the main reasons for such a difference is that soft tissue injury (e.g., laceration) typically occurs without affecting the bone, while hard tissue injury (e.g., bone fracture) happens in a closed space without damaging the skin. Such a difference is largely owing to multiple layers of in-between anatomical structures, such as muscle, fat, or fascia that separate soft and hard tissues.

In the oral cavity, wound healing is unique because bone is situated immediately beneath the oral mucosal tissues and lacks in-between anatomical structures. Consequently, when dentoalveolar trauma such as tooth extraction occurs, the healing of soft tissues (oral mucosa) and hard tissues (alveolar bone) occurs simultaneously [8–10].

Previous studies on osteomucosal healing patterns have been carried out in both humans and animals, mainly focusing in the matters of weeks and months following an injury [11–14]. Although these studies provided valuable insights into tissue remodeling and bone regeneration, critical early events of osteomucosal healing within the first week after extraction have been underexplored. In this study, we hypothesize that there are rapid spatial and temporal changes in soft and hard tissues that are interdependent on each other during osteomucosal healing. To this end, we examined the kinetics of epithelial wound closure, connective tissue formation, and new bone formation, as well as monitored the formation of osteoclasts and the expression of OPG and RANKL at the tooth extraction sites.

## Materials and methods

### Animals

Eight-week-old female C57BL/6 mice (n = 24 mice) were purchased from the Jackson Laboratory (Bar Harbor, ME) and kept in a pathogen-free vivarium in the University of California Los Angeles, Division of Laboratory Animal Medicine. All experiments were performed according to the approved institutional guidelines from the Chancellor's Animal Research Committee (number 2011−062).

## Tooth-extraction mouse model

Tooth extraction was performed under general anesthesia using an intraperitoneal injection of ketamine (100 mg/kg) and xylazine (5 mg/kg). Atraumatic extraction of the maxillary first molars (M1) was carried out using #5 double-end Explorer (Hu-Friedy, Chicago, IL) as described previously [15,16]. Briefly, an explorer was carefully placed between the mesial and distal buccal roots, and gentle buccolingual motions were applied to loosen the tooth. Once sufficient mobility was achieved, the tooth was lifted out of the socket with minimal disruption to the surrounding bone and soft tissues. The extraction sockets were then gently irrigated with a sterile cotton ball to remove excess bleeding, and the animals were monitored postoperatively for signs of distress or complications. Three mice were sacrificed every day for 7 days by cervical dislocation under 1–5% isoflurane in the induction chamber, resulting in a total of 6 extraction sockets per group per time point (2 sockets per mouse × 3 mice per time point). The maxillae were harvested from each mouse and fixed with 4% paraformaldehyde in phosphate-buffered saline (PBS), pH 7.4, at 4°C overnight and stored in 70% ethanol solution.

## µCT Scan

Areas of interest on the maxillae were subjected to µCT scanning (SkyScan1275, Bruker, Kontich, Belgium) at 60 kVp and 166 µA using a voxel size of 10 µm$^3$ and a 0.5 mm Aluminum filter with an integration time of 200 ms using a cylindrical tube (FOV/Diameter: 20.48 mm). The Region of Interest (ROI) was defined as the extraction socket, extending from the most coronal aspect of the alveolar bone crest to the apical portion of the socket. The distal palatal root area was excluded from analyses due to occasional, though rare, occurrences of residual root fragments after extraction. Conversely, distal buccal and mesial roots were consistently and cleanly extracted without residual fragments and therefore included fully in analyses. The ROI was segmented based on grayscale thresholding to differentiate between mineralized and non-mineralized tissues. Two-dimensional slices from each femur were combined using the SkyScan NRecon software to form a three-dimensional reconstruction. The image analysis was performed using the DataViewer and CTAn software.

## Histochemical staining

The scanned tissues were decalcified with 5% ethylenediaminetetraacetic acid (EDTA) and 4% sucrose in PBS, pH 7.4. Decalcification continued for 2–3 weeks at 4°C, and the decalcification solution was changed daily. Fully decalcified tissue samples were sent to the University of California, Los Angeles Translational Procurement Core Laboratory (TPCL) and processed for paraffin embedding. The embedded tissues were sectioned at 5 µm thickness using a microtome. The sectioned slides were deparaffinized at 60°C and then rehydrated in ethanol with an increasing concentration of water. Histologic evaluation of the wound area was performed on decalcified, paraffin-embedded tissue sections stained with hematoxylin and eosin (H&E) using a standard protocol [15]. Tartrate-Resistant Acid Phosphatase (TRAP) staining was performed as described previously [16]. Briefly, EDTA-decalcified tissue sections were deparaffinized at 60°C for 30 minutes. The slides were rehydrated and incubated for 1 hour at 37°C with the TRAP staining solution, according to the manufacturer's protocol (Acid Phosphatase Kit; Sigma-Aldrich, Inc., St. Louis, MO). The slides were counterstained with hematoxylin solution and mounted. Slides were imaged using both brightfield and polarized light microscopy (OLYMPUS U-Pot polarizer, Tokyo, Japan). Wound area and perimeter were measured from histological images using ImageJ. The number of TRAP-positive multinucleated cells within the extraction socket was manually counted under high magnification.

## Trichrome and Picro-Sirius Red staining

Decalcified and paraffin-embedded tissue sections (5 µm thickness) were prepared for collagen staining. Slides were deparaffinized in xylene and rehydrated through a graded ethanol series (100%, 95%, 70%) followed by distilled water.

Trichrome staining was performed using a commercial Trichrome staining kit (Abcam, Cat# ab150686) according to the manufacturer's protocol.

Picrosirius Red staining was conducted to differentiate collagen types based on their birefringence properties. Deparaffinized and rehydrated sections were stained with 0.1% Picrosirius Red solution (Abcam, Cat# ab150681) for 60 minutes at room temperature. Slides were subsequently washed thoroughly in acidified water (0.5% acetic acid), dehydrated in ascending concentrations of ethanol (70%, 95%, and 100%), cleared in xylene, and coverslipped with permanent mounting medium.

Stained slides were imaged under polarized light microscopy (Olympus BX51 equipped with polarized filters). Under polarized light, collagen type I fibers were distinguished by their characteristic yellow-to-red birefringence, whereas collagen type III fibers appeared green. For quantitative analysis, collagen content (types I and III) was measured using the ImageJ software (NIH). Specifically, color threshold segmentation was employed to identify and quantify the birefringent fibers corresponding to collagen types I and III separately. Quantification data were expressed as the percentage of the stained area relative to the total area analyzed, facilitating the assessment of collagen deposition and distribution within the healing sockets.

### Immunohistochemical staining

Deparaffinized sections were immersed in pre-heated sodium citrate buffer for 30 minutes for heat-induced antigen retrieval (HIER). Then, the container with immersed sections was placed on ice and incubated for 30 minutes. After tissue blocking, the tissue sections were incubated overnight at 4°C with primary antibodies for anti-Rankl (C-20, 1:100; Santa Cruz Biotechnology, Inc. Santa Cruz, CA) or anti-OPG (ab183910, 1:100; Abcam, Cambridge, MA). The primary antibodies were detected by incubation with a corresponding biotinylated secondary antibody, an anti-goat IgG or anti-rabbit IgG secondary antibody conjugated with horseradish peroxidase (1:500; Jackson Immuno Research Laboratories, Inc., PA) for 30 minutes at 37°C, and then visualized with 3,3'Diaminobenzidine (DAB) solution (Abcam). Hematoxylin was used for counterstaining.

### Statistical analysis

Changes in parameters over time were analyzed using one-way analysis of variance (ANOVA), followed by Tukey's post hoc test or Sidak for multiple comparisons between time points. All statistical tests were performed using GraphPad Prism 9 (GraphPad Software, San Diego, CA). A p-value of $< 0.05$ was considered statistically significant. Data are presented as mean ± standard deviation.

## Results

### Oral mucosal tissue closure following tooth extraction in mice

To investigate the early events in the osteomucosal healing process, we extracted both maxillary 1st molars (M1) in a mouse and harvested a total of 6 extracted sockets per group per day for 7 days. Tooth extraction resulted in consistent open wounds, with re-epithelialization occurring shortly thereafter (Fig 1A–1C). Histological analysis revealed the presence of a blood clot in the extraction socket immediately after extraction (Fig 1D). By Day 1, the extraction socket was filled with a residual blood clot, transitioning toward the formation of a provisional matrix layer consisting of early inflammatory cell infiltration and initial extracellular matrix deposition. On day 2, the extraction socket was largely filled with a cell-rich provisional matrix, with only a small amount of blood clot remaining. By day 3, roughened bone surfaces became apparent, indicating the onset of bone resorption by osteoclasts. This process led to the rounding of sharp buccal wall margins from day 4. At the same time, epithelial layers began migrating toward the wound, with vascular structures visible in the apical portion. By day 5, fingerlike projections of new bone started to appear from existing bone, accompanied by

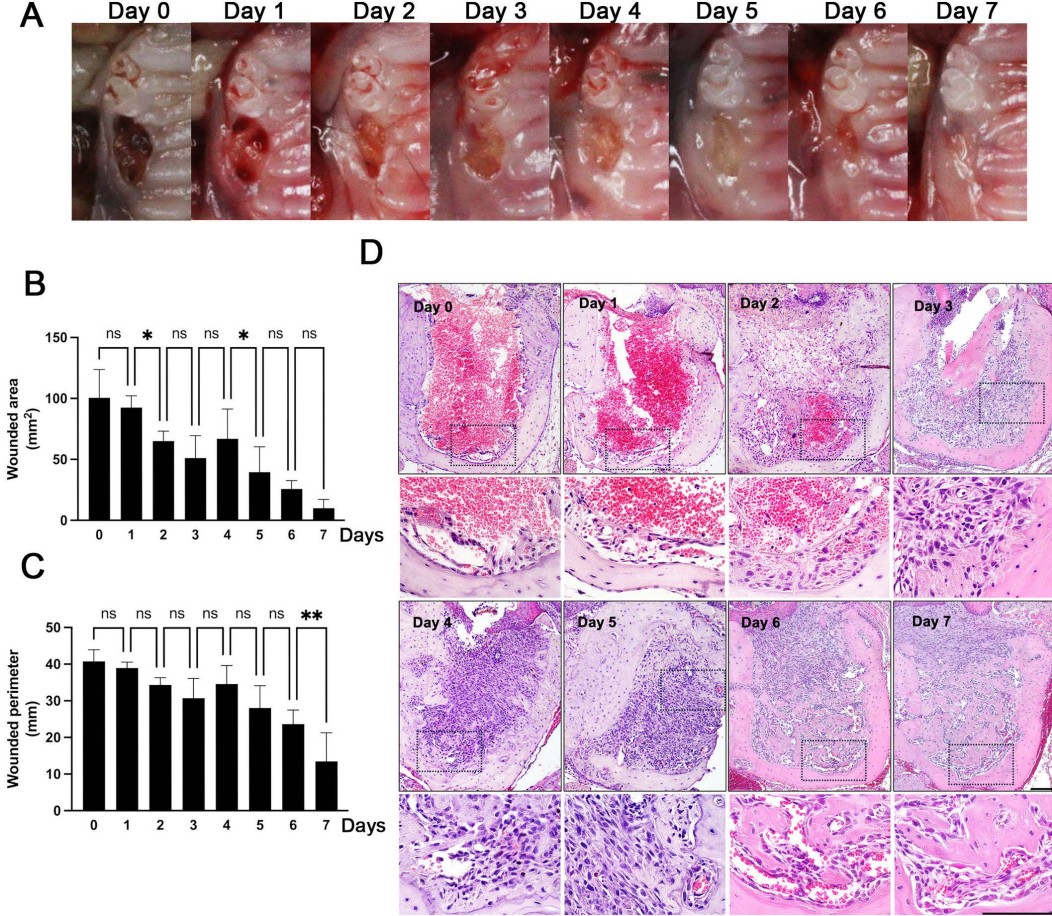

**Fig 1. Oral mucosal tissue closure following tooth extraction in mice. (A)** Representative photographs of tooth-extraction sockets from the maxillary first molar at various post-extraction days. **(B)** Area quantification of the tooth-extraction sockets over the indicated days post-extraction, demonstrating the dynamics of the closure process. **(C)** Perimeter quantification of the tooth-extraction sockets on the indicated days. **(D)** Hematoxylin and eosin (H&E) staining of the tooth-extraction sockets. Scale bars are 100 μm. Square dot boxes indicate the area was magnified for the image below. Graphs show the mean ± SEM. *$p < 0.05$, **$p < 0.005$. ns, not significant. One-way ANOVA with Sidak multiple comparison test **(B and C)**.

isolated bone fragments in the apical and middle portions. By day 6, connective tissue and epithelium completely closed the socket. By day 7, significant amounts of newly forming mineralized tissues were shown in the apical and middle portion of the socket, as well as more fully developed connective tissues and less inflammatory cells, all of which are indicative of proper wound healing after tooth extraction. These daily observations suggest that there is a sequential and coordinated healing process involving simultaneous soft and hard tissue repair after tooth extraction.

## Collagen deposition in the tooth-extraction sockets in mice

Collagen deposition was observed as early as day 2 post-extraction (Fig 2A, top row). Further analysis revealed that both collagen type I (yellow/orange/red) and III (green) followed a similar temporal pattern, appearing on day 2 and increasing exponentially by day 7 (Fig 2A, middle and bottom rows; Fig 2B and 2C). Statistical analyses across time points indicated a significant increase in collagen deposition between day 6 and day 7 ($p < 0.05$), indicating rapid extracellular matrix production during this period.

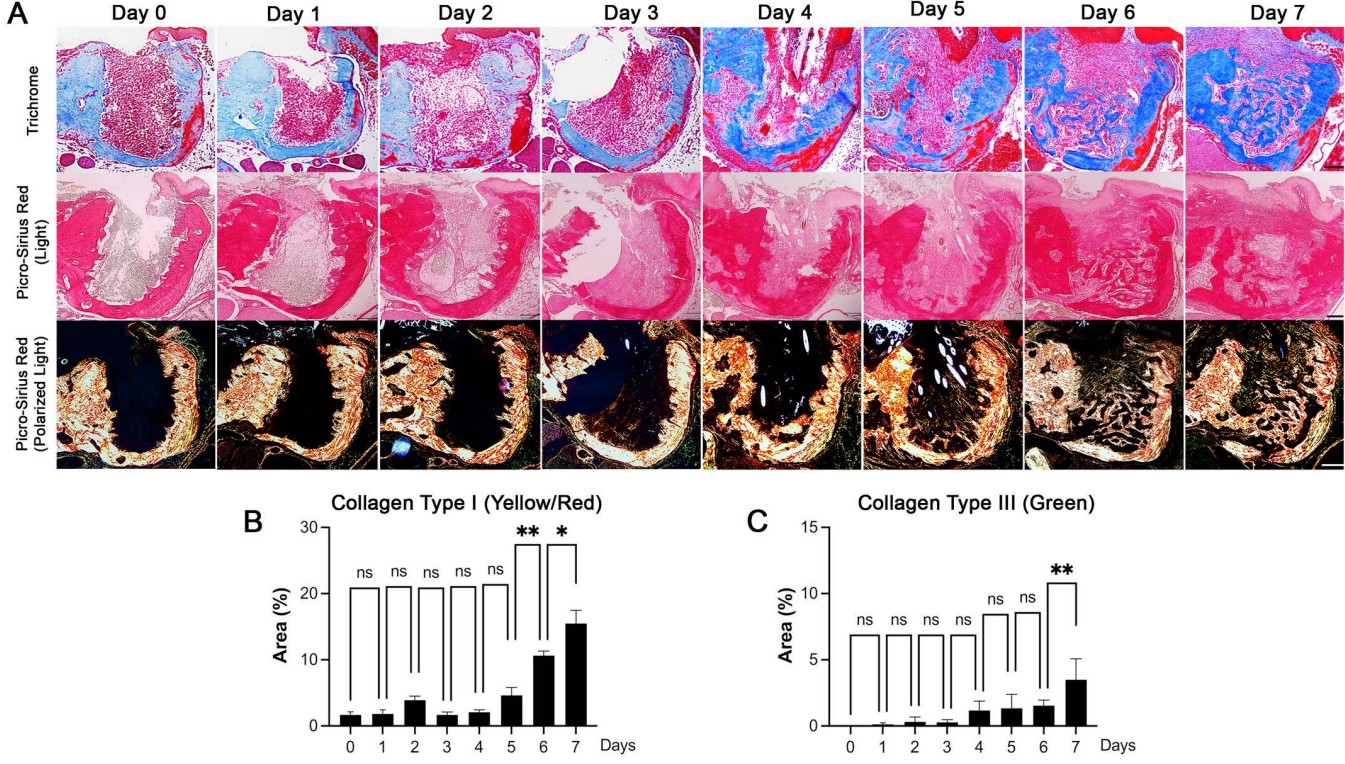

**Fig 2. Collagen deposition in the tooth-extraction sockets in mice. (A)** Trichome (top), Picro-Sirius Red using a light microscope (middle), and Picro-Sirius Red using a polarized light microscope (bottom), staining at the tooth-extraction sockets on the indicated days. **(B)** Quantification of collagen type I (yellow/orange/red) obtained from the Picro-Sirius Red (polarized light) staining at the tooth-extraction sockets. **(C)** Quantification of collagen type III (green) obtained from the Picro-Sirius Red (polarized light) staining in the tooth-extraction sockets. Scale bars are 100 μm. Graphs show the mean ± SEM. *$p < 0.05$, **$p < 0.005$. ns, not significant. One-way ANOVA with Sidak multiple comparison test **(B and C)**.

## Woven bone formation in the tooth-extraction sockets in mice

Longitudinal two-dimensional sagittal μCT images revealed the appearance of radiopaque mineralized foci, indicative of woven bone formation, beginning on Day 5, with substantial expansion by Days 6 and 7 (Fig 3A). Interestingly, 3D reconstructions of the woven bone formation within the socket showed that some of these mineralized tissues are not continuous with the existing bone surface from the socket, suggesting that they formed by *de novo* intramembranous ossification. (Fig 3B and Fig 3B top panel). Quantitative analysis of woven bone volume relative to total socket volume showed a significant increase at Day 7 in both the M and DB root regions (Fig 3C–D). These findings support the interpretation that woven bone is formed de novo within the healing connective tissue and not merely as appositional growth from pre-existing bone surfaces.

## Osteoclast formation in the tooth-extraction sockets in mice

To evaluate the dynamics of osteoclast recruitment during early wound healing, TRAP staining was performed on maxillary extraction sockets. No TRAP-positive osteoclasts were observed immediately after extraction (Day 0). By Day 2, a small number of multinucleated TRAP-positive cells appeared, localized primarily along the edges of pre-existing alveolar bone surfaces (Fig 4A). The number of TRAP+ cells increased significantly between Days 3 and 5, peaking on Day 5, and remained elevated through Day 7 (Fig 4B). Interestingly, osteoclasts were observed exclusively along the edges of

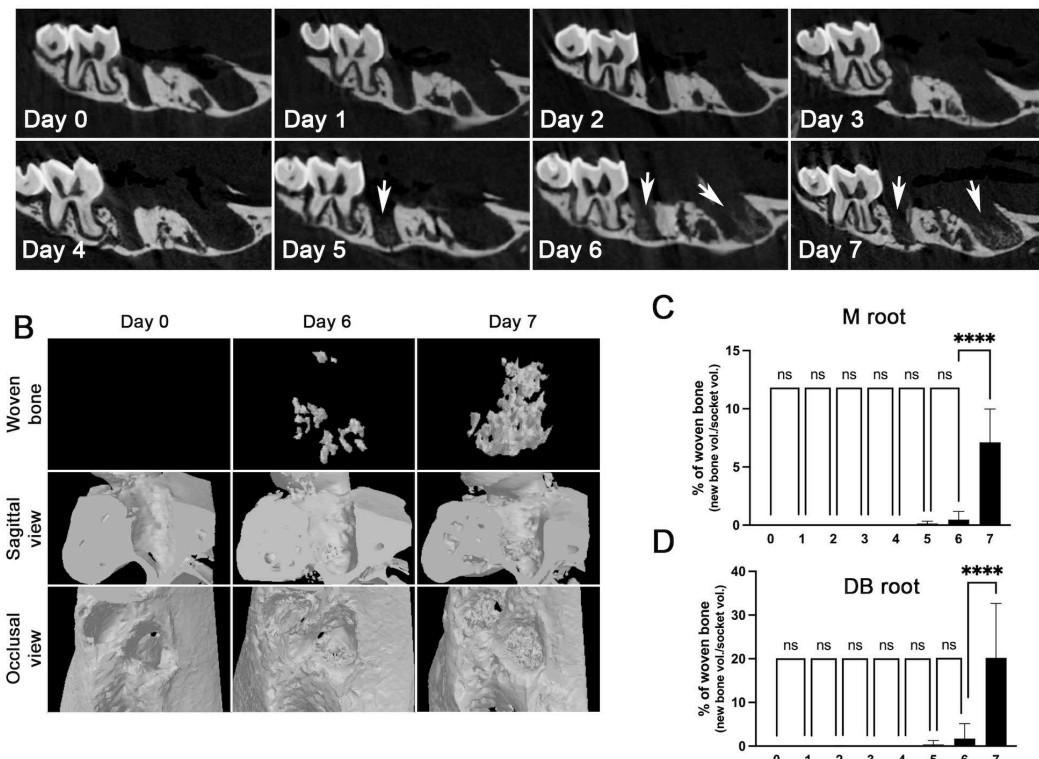

**Fig 3. Woven bone formation in the tooth-extraction sockets in mice. (A)** Two-dimensional cross-sectional views of µCT scanning from the maxillary 1st molar socket. **(B)** Three-dimensional views of µCT scanning at sagittal and occlusal views at the tooth extraction sockets, as well as newly formed woven bone. **(C and D)** Quantification of woven bone in the tooth-extraction sockets at the M and DB roots of the maxillary 1st molar. Graphs show the mean±SEM. ****$p < 0.0001$. ns, not significant. One-way ANOVA with Sidak multiple comparison test **(C and D)**.

pre-existing alveolar bone during the early time points (Days 2–4). However, beginning on Day 5, TRAP-positive cells became detectable on newly formed bone trabeculae when examined under higher magnification (Fig 4A, dashed arrow). These findings suggest that osteoclasts also participate in the early remodeling of de novo bone, in addition to their role in resorbing pre-existing bone margins.

## Expression of RANKL and OPG in the tooth-extraction sockets in mice

Osteoclast differentiation and formation are primarily regulated by RANKL, the osteoclastogenesis inducer, and OPG, the RANKL decoy receptor [17,18]. To examine whether these proteins are associated with bone remodeling, we examined their protein expression using IHC. RANKL expression was observed as early as Day 0, localized intensely at the interface between the clot and the socket wall. This expression persisted throughout the 7-day period, although it became more localized and diminished in intensity by Day 6–7, particularly near the socket periphery (Fig 5A). These findings suggest that RANKL plays a critical role in initiating and sustaining osteoclastogenesis during the early stages of socket healing. On the other hand, OPG expression was low initially but gradually increased starting Day 3, with marked upregulation by Day 5–7 (Fig 5B). OPG immunoreactivity appeared predominantly as diffuse staining, consistent with its function as a secreted protein. Interestingly, OPG expression was predominantly localized around the newly formed woven bone in the extraction sockets. These data suggest that RANKL and OPG regulate osteoclast formation at different spatial and temporal patterns.

**Fig 4. Formation of osteoclasts following tooth extraction in mice. (A)** TRAP staining in the tooth extraction site. **(B)** Quantification of TRAP+ cells. Black boxes indicate the area was magnified for the image below. Arrows denote osteoclasts. Dashed arrows indicate osteoclasts in newly formed bone. Scale bars are 100 µm. Graphs show the mean±SEM. *p<0.05, ****p<0.0001. RM one-way ANOVA with Turkey's multiple comparison test **(B)**.

## Discussion

Although healing after tooth extraction has been studied before, the early events of healing in soft (e.g., epithelial and connective tissues) and hard (e.g., bone tissues) tissues after tooth extraction have not been thoroughly examined. Here, by analyzing the tooth extraction sockets for 1 week daily after tooth extraction in mice, we dissected the orchestration of soft and hard tissues during the early events of the osteomucosal healing, whereby epithelial closure occurs gradually while collagen and osteoclast formation proceeds the woven bone formation (Fig 6). Our study highlighted importance of rapid spatial and temporal changes in the soft and hard tissues that occur sequentially and interdependently in the early stages of the osteomucosal healing process.

During the first week of osteomucosal healing, collagen formation was one of the earliest structural changes seen in the extraction socket (Fig 2). Epithelial cells progressively migrated toward the tooth extraction socket onto the collagen matrices that rapidly occur after blood clot formation. This observation is in line with previous findings; in cutaneous wound healing, collagen beds produced by fibroblasts are essential for the epithelial cells to migrate and induce wound closure [19]. Similarly, in bone healing, one of the early steps in bone formation is the osteoblast-mediated secretion of osteoid, which is an uncalcified collagen matrix that eventually becomes mineralized upon calcification

**A**

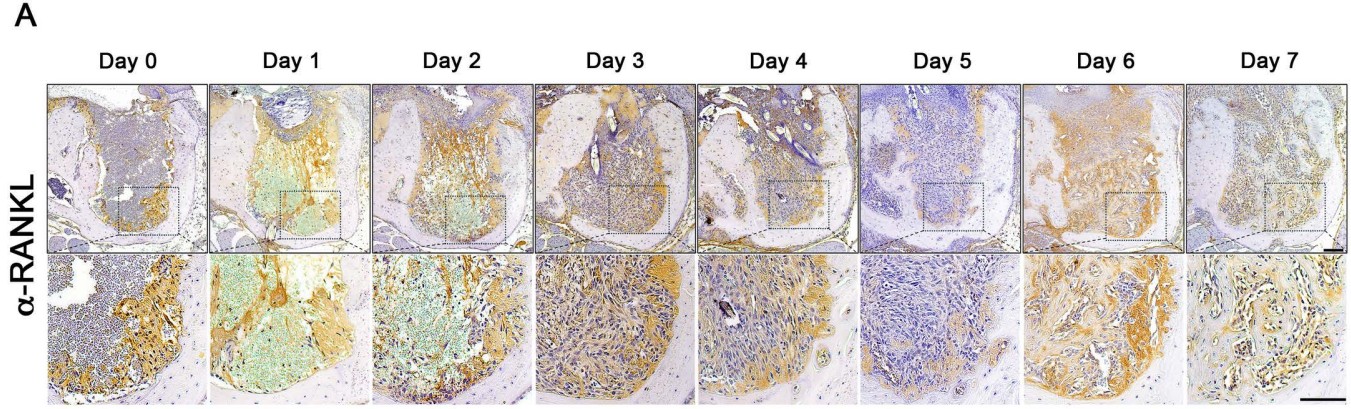

**B**

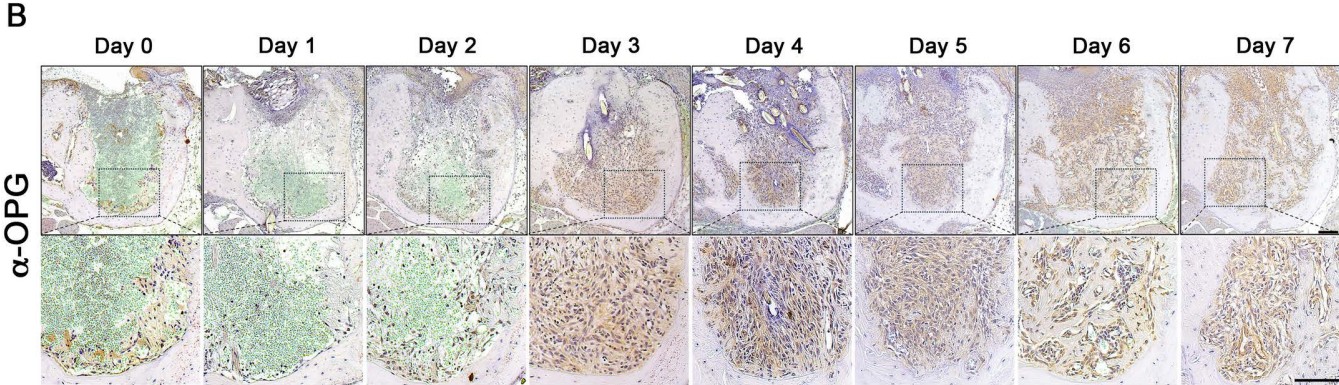

**Fig 5. Expression of RANKL and OPG in the tooth-extraction sockets in mice.** Immunohistochemical staining was performed against RANKL (A) or OPG (B) in the tooth-extraction sockets. Square dot boxes indicate the area was magnified for the image below. Scale bars are 100 μm.

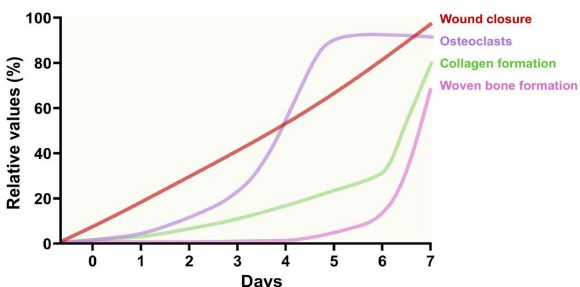

**Fig 6. Summary of osteomucosal healing patterns.** Wound closure occurs progressively in a time-dependent manner (red). The presence of osteoclasts (purple) and the formation of collagen (green) fibers occur as early as day 2. Newly forming woven bone (pink) starts forming at day 5 and increases rapidly at day 7.

[20]. As such, the formation of collagen matrix is the required prerequisite event during osteomucosal healing at the tooth-extraction socket, functioning as a "natural" scaffold onto which epithelial and bone cells get recruited for proper soft and hard tissue healing.

We observed rapid woven bone formation in the extraction socket, with significant development by day 7 post-tooth extraction (Fig 3). The presence of woven bone in the socket as early as day 7 has been noted in previous studies with mice, although the rapidity of this presence was less appreciated [21,22]. Bone mineralization is a non-linear process involving an initial osteoid formation phase, followed by a rapid mineralization phase that achieves 75% of the total expected mineralization within a couple of days [23,24]. Therefore, bone formation in the socket post-extraction does not occur gradually but rather exponentially only after certain conditions are met, such as collagen formation and proper wound closure (Fig 6).

After tooth extraction, osteoclasts began to appear on the existing bone surfaces as early as day 2 and progressively increased up to day 6, after which the number of osteoclasts was maintained. A similar experiment performed by Vieira *et al.* also found osteoclasts in the first week after tooth extraction, although the number was low [12]. This discrepancy may be attributed to the employment of different methods for osteoclast identification (e.g., histological identification vs. TRAP staining). Nonetheless, it is worth noting that the early presence of osteoclasts allows for bone resorption at the existing bone surface and triggers osteoclast-osteoblast coupling to stimulate new bone formation [25]. Indeed, our study showed that the maximum number of osteoclasts coincided with new bone formation on the surface of the existing bone (Fig 3 and 4), indicating that the surface of the existing bone in the tooth-extraction socket is the active site of the osteoclast-osteoblast coupling.

Interestingly, we also observed islands of mineralized nodules in the socket that were disconnected from the newly formed bone that arose from the existing surface of the bone socket (Fig 2A, Picro-Sirius Red staining; Fig 3B, top panel). *De novo* bone formation apart from the existing bone surface has been observed during osteointegration of the endosseous implants [26]. Therefore, these observations indicate that new bone formation in the tooth-extraction socket occurs in two ways: 1) osteoclast-dependent bone formation from the existing bone surface via osteoclast-osteoblast coupling, and 2) osteoclast-independent *de novo* bone formation via intramembranous ossification.

These observations have several important clinical implications. First, tooth extraction often leads to alveolar osteitis (AO), also known as "dry socket." AO commonly occurs when the blood clots become partially or totally dislodged from the socket, and healing of the dry socket may last several weeks [27]. Based on our study, it is possible that the lodged blood clots may remove an opportunity to induce *de novo* bone formation in an osteoclast-independent manner, leaving the bone formation mode only from the existing bone surfaces that is osteoclast-dependent, a process of which can be slow and further delays the proper osteomucosal wound healing process. Second, in patients taking anti-resorptive agents such as bisphosphonates or denosumab, the osteoclast-osteoblast coupling mechanism is compromised due to inhibition of osteoclast formation and/or functions [28]. As such, bone formation in the socket is primarily dependent on *de novo* bone formation in the blood clots. In these patients, if the blood clots become dislodged after tooth extraction, or if bleeding does not get induced, it may lead to medication-related osteonecrosis of the jaw (MRONJ). Lastly, wound healing in the oral cavity is known for expedited healing with minimal cicatrization, and this scar formation is primarily mediated by dysregulated production of collagens by fibroblasts [2]. Recent studies suggest that microanatomic differences of fibroblasts lead to scar formation; fibroblasts residing in the deep layers are more prone to cause scar formation when compared to those in the superficial layers [29]. As such, it is possible to speculate that deeper fibroblasts may contribute more to bone formation, rather than scar formation, that leads to scarless healing unique to the oral cavity. Further studies await confirmations of these interpretations.

It is worth noting that TRAP+ osteoclasts were initially localized predominantly on the surface of pre-existing alveolar bone during the early healing phase (Days 2–4). However, on Day 5, osteoclasts were also observed on newly formed woven bone trabeculae, as confirmed under higher magnification (Fig 4A). This spatial shift suggests a transition in osteoclast function—from remodeling existing bone to engaging in the early remodeling of de novo bone structures within the healing socket.

However, when we examined the expression level of RANKL and OPG, two essential regulators of osteoclast formation [17,18], we observed that the level of OPG was dramatically increased within the tooth-extraction sockets around the newly formed bone (Fig 5B). As such, it is conceivable that increased expression of OPG allows for inhibition of osteoclast formation around the newly forming bone areas. The precise mechanisms underlying this preferential bone resorptive function of osteoclasts remain to be elucidated.

This study provides important insights about the cellular events during the early stages of osteomucosal healing; however, it has several limitations. First, this study is a pilot in nature with 6 extraction sites per group (n = 3 mice). Although our study conferred meaningful results with statistically significant differences, more robust analysis with increased numbers of mice and extraction sockets may be needed to generalize the findings. Second, it is worth noting that there are differences in healing rates, tissue composition, and immune responses between rodents and humans. Although there is no universal "conversion rate" between rodent and human pathophysiological processes [30], healing in rodents occurs relatively faster than that in humans [31]. Therefore, species differences must be considered when extrapolating these results to clinical scenarios. Lastly, our study focused on cellular changes during osteomucosal healing. Behaviors of the cells involved in the wound healing processes are primarily governed by molecular cues from their microenvironments. As such, additional molecular studies should be followed to support the findings observed in this study.

In conclusion, our study provides the first detailed characterization of the early cellular events in osteomucosal healing following tooth extraction in mice. We identify collagen deposition, osteoclast activity, and woven bone formation as sequential but overlapping processes that drive simultaneous soft and hard tissue repair. These findings advance our understanding of the mechanisms underlying uneventful healing in the oral cavity and provide a foundation for future research into clinical interventions for optimizing wound healing outcomes.

## Supporting information

**S1 Table. Wound area and perimeter for Fig. 1.**
(XLSX)

**S2 Table. Collagen I and III measurement for Fig 2.**
(XLSX)

**S3 Table. Woven bone measurement for Fig 3.**
(XLSX)

**S4 Table. TRAP+ measurement for Fig 4.**
(XLSX)

## Author contributions

**Conceptualization:** Sol Kim, Minju Song, Drake Williams, Ki-Hyuk Shin, Reuben H. Kim.

**Data curation:** Sol Kim, Minju Song, Drake Williams, Wen Du, Inwoo Cho, Joey Kim, Ahana Goswami, Reuben H. Kim.

**Formal analysis:** Sol Kim, Minju Song, Drake Williams, Wen Du, Inwoo Cho, Eunbin Bae, Joey Kim, Ahana Goswami, Ki-Hyuk Shin, Reuben H. Kim.

**Funding acquisition:** Reuben H. Kim.

**Investigation:** Sol Kim, Minju Song, Drake Williams, Inwoo Cho, Joey Kim, Ahana Goswami, Reuben H. Kim.

**Methodology:** Ki-Hyuk Shin, No-Hee Park, Reuben H. Kim.

**Project administration:** Eunbin Bae, Reuben H. Kim.

**Resources:** No-Hee Park, Reuben H. Kim.

**Software:** Reuben H. Kim.

**Supervision:** Sol Kim, Ki-Hyuk Shin, Reuben H. Kim.

**Validation:** Sol Kim, Reuben H. Kim.

**Visualization:** No-Hee Park, Reuben H. Kim.

**Writing – original draft:** Sol Kim, Eunbin Bae, Reuben H. Kim.

**Writing – review & editing:** Sol Kim, Minju Song, Drake Williams, Wen Du, Inwoo Cho, Eunbin Bae, Joey Kim, Ahana Goswami, Ki-Hyuk Shin, No-Hee Park, Reuben H. Kim.

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
