## [Decision Letter · Decision Letter 0]

26 Dec 2024

PONE-D-24-55861Early cellular and molecular events of osteomucosal healing in the tooth extraction socketPLOS ONE

Dear Dr. Kim,

Thank you for submitting your manuscript to PLOS ONE. After careful consideration, we feel that it has merit but does not fully meet PLOS ONE’s publication criteria as it currently stands. Therefore, we invite you to submit a revised version of the manuscript that addresses the points raised during the review process.

We look forward to receiving your revised manuscript.

Regards,

Henrique Hadad, DDS, Ph.D.   

Academic Editor

PLOS ONE

“This study was supported in part by grants from NIH/NIDCR DE023348 and DE033281 (RHK). The authors declare no potential conflicts of interest with respect to the authorship and/or publication of this article.”

6. Please amend either the abstract on the online submission form (via Edit Submission) or the abstract in the manuscript so that they are identical.

Additional Editor Comments:

Dear Dr. Kim,

Thank you for submitting your manuscript to PLOS One. Your paper is undoubtedly interesting and raises important points that I believe will be highly appreciated in the field. However, the majority of the reviewers have raised some significant concerns about the paper that need to be addressed before it can be considered for publication.

My considerations focus on the Methods section, addressing specific points such as:

- The authors did not present the ARRIVE guidelines.

- The sample size was not justified.

- Details regarding histochemical staining need clarification—specifically, describe the TRAP immunohistochemistry.

- Some methodology is repeated in the Results section.

- Trichrome staining is not included in the Methods section. (Also, which type of Trichrome staining was used? Masson's?)

- Please explain how osteoclast coating was performed.

- Lastly, please include a clear discussion of the limitations of your study.

I look forward to receiving your revised manuscript.

Best regards,

Dr Henrique Hadad, DDS, PhD

Academic Editor - PLOS One

Reviewers' comments:

Reviewer #1:

This is a nice study with an easy-to-read flow, that gathers attention from the reader.

Minor questions and suggestions should be assessed by the authors:

1) When the authors cite MRONJ in the discussion panel, it should be discussed with another statement " The early presence of osteoclasts allows for bone resorption at the existing bone surface and triggers osteoclast-osteoblast coupling to stimulate new bone formation".

2) The authors should discuss the timing of socket cicatrization. It is interesting and not well addressed in the literature how soft tissue and hard tissue heal at different paces but a similar finish line.

3) This is a great study, with interesting results. Image 06 does not represent this study, but a grade school paperwork. The authors should present a professional Graphic for Image 06.

Reviewer #2:

Comments to the author

The manuscript is focused in evaluating epithelial regeneration, connective tissue proliferation and bone formation over the first week following tooth extraction in mice. The authors claim this is unprecedented data, however, the cascade of events occurring during socket healing is well documented. Maybe, their design (consecutive euthanasia over the week) might have added information, but the authors failed to make clear which gap was filled by their data.

Besides this limitation that should be addressed, some other major changes need to be done prior to re-consideration for publication, once the authors succeed in

ABASTRACT

Use the background to state the singularity of osteomucosal healing and exactly which information is been added by your research. “Early events” is vague.

INTRODUCTION

The rationale is currently based in distinct healing across tissues. Instead, the introduction should focus on the unknown events in osteomucosal healing and plausible crosstalk between the events occurring within the socket.

MATERIALS AND METHODS

Did you perform sample size calculation? Based on most literature of socket healing in mice, N=3 seems not sufficient to assure relevance.

You mention that both right and left first molars were extracted, which is different from the abstract. Which one is correct?

Describe how you defined the region of interest and the method of analysis for the microCT.

Mason’s Trichrome was only mentioned in the results and the information is lacking in the M&M.

The method of analysis for all outcome parameters was neglected or poorly described. An extensive revision on that is necessary.

RESULTS

The clinical analysis should focus on wound closure only, instead of describing tissues. Epithelium can be appreciated in the histological slides.

The word “matured” should be changed to proper histologic terminology.

The figures are shown in poor quality, hindering a proper appreciation of their representativeness. Additionally, they are presented in low magnification, which compromises a proper visualization of the tissue events described in the text.

The IHC is also shown in low magnification. I understand that the labeling is not confined to the cytosolic compartment since these proteins exert their function when secreted in the extracellular matrix. However, higher magnification images are necessary to identify what is antigen-antibody labeling and what is background labeling.

Discussion

Overall, the discussion is missing comprehensive understanding of the data shown by the authors. Also, the authors could explore the sequential (and overlapping) events trying to establish a spatial relevance of uneventful osteomucosal healing.

Conclusion

A proper conclusion based on the results should be drawn.

Reviewer #3:

The proposed paper details the first week of alveolar repair in daily analyses. A very interesting study that will certainly support studies regarding interferences in the natural course.

The objectives are clear and the methodology is consolidated in the literature in relation to alveolar repair.

It precisely details the role of osteoclasts in the early stages after tooth extraction, which is well explored in the discussion as possible indicators for studies of the mechanisms of MRONJ.

Based on the analysis of the study, I suggest some small changes:

1. There is a tendency in the results section to repeat some points from the methods, for example:

- “To investigate the kinetics of collagen formation and deposition following tooth extraction, we stained the sections with Trichrome, a maker of collagens in general…”

- “To investigate the kinetics of collagen formation and deposition following tooth extraction, we stained the sections with Trichrome, a maker of collagens in general..”

I suggest that these points be part of the method only and not repeated in the results section.

2. Regarding Figure 6: although it seems obvious that the markings on the horizontal axis refer to the 7 days studied, I suggest that the numbers of the days to which the markings refer be written. In the same way, markings be added to the vertical axis.

3. In the caption of Figure 1, I suggest removing "N = 6 tooth

extraction sites with 3 mice per group.", which refers to the method and is already very clear in the text.

Reviewer #4:

Kim et al. presented an original study investigating early cellular and molecular events in osteomucosal healing in a mouse model. This study provides a new understanding of collagen deposition, osteoclast activity, and remodeling. The results have clear clinical implications for MRONJ and dry sockets using advanced imaging and staining techniques. Overall, the manuscript provides valuable contributions to the understanding of osteomucosal healing. Its clarity and impact will be further enhanced by addressing the following suggestions.

Abstract: While the abstract is informative, it lacks brief details on the materials and methods used. Including this information would improve clarity and help readers understand how the results were obtained.

Suggestions for improvement:

1. Although the figures and results are robust, the discussion could be more explicit in linking the results to potential future studies on clinical implications, such as MRONJ or dry socket.

2. The narrative would be streamlined, and readability would be improved by reducing redundancy in the introduction and discussion.

3. This study did not use additional molecular methods, such as qRT-PCR for gene expression analysis, that could have deepened the findings.

4. Potential limitations, such as differences in healing mechanisms between mice and humans, should be clarified. This would better contextualize the findings.

Reviewer #5:

I was pleased to review this manuscript. The initial stages of healing in fresh sockets remain highly important and intriguing. However, the paper lacks adequate support from the existing literature, despite the availability of a substantial works on the topic that could better support the study.

I recommend major review based on the following comments:

Abstract: I suggest restructuring the abstract, as the study design is not well described, nor are the results and conclusion clearly presented. Additionally, consider adding keywords to improve the visibility of your study in database searches.

Introduction: The introduction lacks sufficient references to support the context and rationale of the study. While the aim is stated, there is no clear hypothesis. What did the authors expect to find or address within this gap?

Methods: Methodologically, this is essentially a result based on histomorphometric parameters, as you did not explore deeply into gene and protein expression in the samples. It is important to strengthen what is being presented. Alternatively, you could change the approach to frame it as a pilot study investigating the initial events of alveolar remodeling.

Did the authors perform a sample size calculation? Considering that the number of animals per time point is very low, with only three animals per period, this would not provide sufficient statistical confidence, even with the use of a split-mouth design. doi: 10.4103/0976-500X.119726

"Tooth-extraction mouse model" : What instrument was used for the atraumatic extraction? If your article is published, it must be reproducible, allowing other researchers to replicate the study. Additionally, others may draw inspiration from your work for future studies.

μCT Scan: The parameters evaluated are not described, nor are they presented in the results. I suggest using parameters from the literature, such as those described in the study by Viera (2015), as cited by the authors.

Histochemical Staining: Improve the description, there is no cited eosin staining with hematoxilin.

Immunohistochemical staining: Why did the authors use only OPG and RANKL markers? In the early stages, there are many other important markers that could have been investigated, such as osteopontin, osteocalcin, BSP, and Runx2.

Statistical Analysis: What do the authors mean by "To compare the group differences" if only one group is being analyzed at different time points? Why were so many significance factors used? Since you mentioned that the p-value is < 0.05, the other p-values shown are already within this range.

RESULTS: In general, the results are not well presented, especially in the description. They are confusing, and it is not possible to correlate them with what is being shown in the images, which also lack sufficient resolution for the message the author intends to convey to the readers. I suggest using images with higher magnification.

For example: "By day 3, roughened bone surfaces were observed in the tooth-extracted sockets, suggesting that bone remodeling had started through bone resorption. As a result of this bone remodeling, the sharp edge of the buccal wall margin of the extraction socket became rounded by day 4." This cannot be visualized properly. I suggest using arrows or asterisks to highlight what you would like to show in the images. Once again, the study cited in your references by Vieira is well illustrated in terms of the images.

Woven bone formation: You have access to a high-resolution 3D imaging tool, but it is not being used to its full potential. I suggest utilizing better evaluation parameters, as it was not described how the region of interest (ROI) was selected. Once again, I recommend referring to the studies by Vieira and Bouxsein (2010) DOI: 10.1002/jbmr.141 for better guidance.

Osteoclast formation: I am unable to visualize the osteoclasts present. While we know how the histological sections were made and their positioning, I suggest providing images with higher magnification specifically of the representative areas where these cell types need to be visualized.

Discussion: The discussion is very bold and poorly supported by the literature. Many statements are not backed by existing references. Additionally, it is confusing, as it moves back and forth on the same topic across several paragraphs. I suggest a thorough rereading and restructuring to improve clarity and coherence.

"Therefore, these observations indicate that new bone formation in the tooth-extraction socket occurs in two ways: 1) osteoclast-dependent bone formation from the existing bone surface via osteoclast-osteoblast coupling, and 2) osteoclast-independent de novo bone formation via intramembranous ossification." - How can you strongly affirm this if your sample size is low, and you have only evaluated histomorphometric parameters? Such imply may require more robust data, including gene and protein expression analysis, and a larger sample size to provide stronger evidence.

"The reason for this preferential presence of osteoclasts is presently unclear; however, it seems intuitive that nature has evolved to prevent osteoclasts from targeting the newly forming bone during the healing process to increase the efficiency of new bone formation." - If it is unclear, how can you suggest this information? Based on what evidence or reasoning is this claim being made? It is important to provide solid references or data to support such hypotheses, especially when the mechanism is not yet fully understood.

The conclusion is not clear, and the text ends abruptly with a paragraph that seems to suggest continuity, followed by another strong, unreferenced statement: "Unlike other parts of the body, the healing of the soft and hard tissues in the oral cavity often occurs simultaneously after a traumatic event such as tooth extraction." A conclusion needs to be well-defined and should summarize the key findings in a clear and coherent manner, avoiding unsupported or vague statements.

Reviewers' comments:

Reviewer's Responses to Questions

**Comments to the Author**

1. Is the manuscript technically sound, and do the data support the conclusions?

Reviewer #1: Yes

Reviewer #2: Partly

Reviewer #3: Yes

Reviewer #4: Yes

Reviewer #5: Partly

2. Has the statistical analysis been performed appropriately and rigorously? 

Reviewer #1: Yes

Reviewer #2: I Don't Know

Reviewer #3: Yes

Reviewer #4: Yes

Reviewer #5: No

3. Have the authors made all data underlying the findings in their manuscript fully available?

Reviewer #1: Yes

Reviewer #2: No

Reviewer #3: Yes

Reviewer #4: Yes

Reviewer #5: Yes

4. Is the manuscript presented in an intelligible fashion and written in standard English?

Reviewer #1: Yes

Reviewer #2: Yes

Reviewer #3: Yes

Reviewer #4: Yes

Reviewer #5: Yes

5. Review Comments to the Author

Reviewer #1: This is a nice study with an easy-to-read flow, that gathers attention from the reader.

Minor questions and suggestions should be assessed by the authors:

1) When the authors cite MRONJ in the discussion panel, it should be discussed with another statement " The early presence of osteoclasts allows for bone resorption at the existing bone surface and triggers osteoclast-osteoblast coupling to stimulate new bone formation".

2) The authors should discuss the timing of socket cicatrization. It is interesting and not well addressed in the literature how soft tissue and hard tissue heal at different paces but a similar finish line.

3) This is a great study, with interesting results. Image 06 does not represent this study, but a grade school paperwork. The authors should present a professional Graphic for Image 06.

Reviewer #2: Comments to the author

The manuscript is focused in evaluating epithelial regeneration, connective tissue proliferation and bone formation over the first week following tooth extraction in mice. The authors claim this is unprecedented data, however, the cascade of events occurring during socket healing is well documented. Maybe, their design (consecutive euthanasia over the week) might have added information, but the authors failed to make clear which gap was filled by their data.

Besides this limitation that should be addressed, some other major changes need to be done prior to re-consideration for publication, once the authors succeed in

ABASTRACT

Use the background to state the singularity of osteomucosal healing and exactly which information is been added by your research. “Early events” is vague.

INTRODUCTION

The rationale is currently based in distinct healing across tissues. Instead, the introduction should focus on the unknown events in osteomucosal healing and plausible crosstalk between the events occurring within the socket.

MATERIALS AND METHODS

Did you perform sample size calculation? Based on most literature of socket healing in mice, N=3 seems not sufficient to assure relevance.

You mention that both right and left first molars were extracted, which is different from the abstract. Which one is correct?

Describe how you defined the region of interest and the method of analysis for the microCT.

Mason’s Trichrome was only mentioned in the results and the information is lacking in the M&M.

The method of analysis for all outcome parameters was neglected or poorly described. An extensive revision on that is necessary.

RESULTS

The clinical analysis should focus on wound closure only, instead of describing tissues. Epithelium can be appreciated in the histological slides.

The word “matured” should be changed to proper histologic terminology.

The figures are shown in poor quality, hindering a proper appreciation of their representativeness. Additionally, they are presented in low magnification, which compromises a proper visualization of the tissue events described in the text.

The IHC is also shown in low magnification. I understand that the labeling is not confined to the cytosolic compartment since these proteins exert their function when secreted in the extracellular matrix. However, higher magnification images are necessary to identify what is antigen-antibody labeling and what is background labeling.

Discussion

Overall, the discussion is missing comprehensive understanding of the data shown by the authors. Also, the authors could explore the sequential (and overlapping) events trying to establish a spatial relevance of uneventful osteomucosal healing.

Conclusion

A proper conclusion based on the results should be drawn.

Reviewer #3: The proposed paper details the first week of alveolar repair in daily analyses. A very interesting study that will certainly support studies regarding interferences in the natural course.

The objectives are clear and the methodology is consolidated in the literature in relation to alveolar repair.

It precisely details the role of osteoclasts in the early stages after tooth extraction, which is well explored in the discussion as possible indicators for studies of the mechanisms of MRONJ.

Based on the analysis of the study, I suggest some small changes:

1. There is a tendency in the results section to repeat some points from the methods, for example:

- “To investigate the kinetics of collagen formation and deposition following tooth extraction, we stained the sections with Trichrome, a maker of collagens in general…”

- “To investigate the kinetics of collagen formation and deposition following tooth extraction, we stained the sections with Trichrome, a maker of collagens in general..”

I suggest that these points be part of the method only and not repeated in the results section.

2. Regarding Figure 6: although it seems obvious that the markings on the horizontal axis refer to the 7 days studied, I suggest that the numbers of the days to which the markings refer be written. In the same way, markings be added to the vertical axis.

3. In the caption of Figure 1, I suggest removing "N = 6 tooth

extraction sites with 3 mice per group.", which refers to the method and is already very clear in the text.

Reviewer #4: Kim et al. presented an original study investigating early cellular and molecular events in osteomucosal healing in a mouse model. This study provides a new understanding of collagen deposition, osteoclast activity, and remodeling. The results have clear clinical implications for MRONJ and dry sockets using advanced imaging and staining techniques. Overall, the manuscript provides valuable contributions to the understanding of osteomucosal healing. Its clarity and impact will be further enhanced by addressing the following suggestions.

Abstract: While the abstract is informative, it lacks brief details on the materials and methods used. Including this information would improve clarity and help readers understand how the results were obtained.

Suggestions for improvement:

1. Although the figures and results are robust, the discussion could be more explicit in linking the results to potential future studies on clinical implications, such as MRONJ or dry socket.

2. The narrative would be streamlined, and readability would be improved by reducing redundancy in the introduction and discussion.

3. This study did not use additional molecular methods, such as qRT-PCR for gene expression analysis, that could have deepened the findings.

4. Potential limitations, such as differences in healing mechanisms between mice and humans, should be clarified. This would better contextualize the findings.

Reviewer #5: I was pleased to review this manuscript. The initial stages of healing in fresh sockets remain highly important and intriguing. However, the paper lacks adequate support from the existing literature, despite the availability of a substantial works on the topic that could better support the study.

I recommend major review based on the following comments:

Abstract: I suggest restructuring the abstract, as the study design is not well described, nor are the results and conclusion clearly presented. Additionally, consider adding keywords to improve the visibility of your study in database searches.

Introduction: The introduction lacks sufficient references to support the context and rationale of the study. While the aim is stated, there is no clear hypothesis. What did the authors expect to find or address within this gap?

Methods: Methodologically, this is essentially a result based on histomorphometric parameters, as you did not explore deeply into gene and protein expression in the samples. It is important to strengthen what is being presented. Alternatively, you could change the approach to frame it as a pilot study investigating the initial events of alveolar remodeling.

Did the authors perform a sample size calculation? Considering that the number of animals per time point is very low, with only three animals per period, this would not provide sufficient statistical confidence, even with the use of a split-mouth design. doi: 10.4103/0976-500X.119726

"Tooth-extraction mouse model" : What instrument was used for the atraumatic extraction? If your article is published, it must be reproducible, allowing other researchers to replicate the study. Additionally, others may draw inspiration from your work for future studies.

μCT Scan: The parameters evaluated are not described, nor are they presented in the results. I suggest using parameters from the literature, such as those described in the study by Viera (2015), as cited by the authors.

Histochemical Staining: Improve the description, there is no cited eosin staining with hematoxilin.

Immunohistochemical staining: Why did the authors use only OPG and RANKL markers? In the early stages, there are many other important markers that could have been investigated, such as osteopontin, osteocalcin, BSP, and Runx2.

Statistical Analysis: What do the authors mean by "To compare the group differences" if only one group is being analyzed at different time points? Why were so many significance factors used? Since you mentioned that the p-value is < 0.05, the other p-values shown are already within this range.

RESULTS: In general, the results are not well presented, especially in the description. They are confusing, and it is not possible to correlate them with what is being shown in the images, which also lack sufficient resolution for the message the author intends to convey to the readers. I suggest using images with higher magnification.

For example: "By day 3, roughened bone surfaces were observed in the tooth-extracted sockets, suggesting that bone remodeling had started through bone resorption. As a result of this bone remodeling, the sharp edge of the buccal wall margin of the extraction socket became rounded by day 4." This cannot be visualized properly. I suggest using arrows or asterisks to highlight what you would like to show in the images. Once again, the study cited in your references by Vieira is well illustrated in terms of the images.

Woven bone formation: You have access to a high-resolution 3D imaging tool, but it is not being used to its full potential. I suggest utilizing better evaluation parameters, as it was not described how the region of interest (ROI) was selected. Once again, I recommend referring to the studies by Vieira and Bouxsein (2010) DOI: 10.1002/jbmr.141 for better guidance.

Osteoclast formation: I am unable to visualize the osteoclasts present. While we know how the histological sections were made and their positioning, I suggest providing images with higher magnification specifically of the representative areas where these cell types need to be visualized.

Discussion: The discussion is very bold and poorly supported by the literature. Many statements are not backed by existing references. Additionally, it is confusing, as it moves back and forth on the same topic across several paragraphs. I suggest a thorough rereading and restructuring to improve clarity and coherence.

"Therefore, these observations indicate that new bone formation in the tooth-extraction socket occurs in two ways: 1) osteoclast-dependent bone formation from the existing bone surface via osteoclast-osteoblast coupling, and 2) osteoclast-independent de novo bone formation via intramembranous ossification." - How can you strongly affirm this if your sample size is low, and you have only evaluated histomorphometric parameters? Such imply may require more robust data, including gene and protein expression analysis, and a larger sample size to provide stronger evidence.

"The reason for this preferential presence of osteoclasts is presently unclear; however, it seems intuitive that nature has evolved to prevent osteoclasts from targeting the newly forming bone during the healing process to increase the efficiency of new bone formation." - If it is unclear, how can you suggest this information? Based on what evidence or reasoning is this claim being made? It is important to provide solid references or data to support such hypotheses, especially when the mechanism is not yet fully understood.

The conclusion is not clear, and the text ends abruptly with a paragraph that seems to suggest continuity, followed by another strong, unreferenced statement: "Unlike other parts of the body, the healing of the soft and hard tissues in the oral cavity often occurs simultaneously after a traumatic event such as tooth extraction." A conclusion needs to be well-defined and should summarize the key findings in a clear and coherent manner, avoiding unsupported or vague statements.

6. PLOS authors have the option to publish the peer review history of their article (what does this mean?). If published, this will include your full peer review and any attached files.

Reviewer #1: **Yes: **Samuel Macedo Costa

Reviewer #2: No

Reviewer #3: No

Reviewer #4: **Yes: **Avneesh Chopra

Reviewer #5: No

---

## [Author Response · Author response to Decision Letter 1]

14 Mar 2025

Responses to Reviewers

We deeply appreciate the referees’ comments. We prepared our point-by-point responses (below in blue) and modified the manuscript to correctly address all their concerns. Please see below for our answers.

REVIEWER #1:

This is a nice study with an easy-to-read flow, that gathers attention from the reader. Minor questions and suggestions should be assessed by the authors:

1.1) When the authors cite MRONJ in the discussion panel, it should be discussed with another statement " The early presence of osteoclasts allows for bone resorption at the existing bone surface and triggers osteoclast-osteoblast coupling to stimulate new bone formation".

Our response: Thank you for your suggestion. We have included a paragraph dedicating for the clinical implications of our study. In there, we discussed MRONJ more in depth as the following.

1.2) The authors should discuss the timing of socket cicatrization. It is interesting and not well addressed in the literature how soft tissue and hard tissue heal at different paces but a similar finish line.

Our response: Thank you for your insightful comment. We now have included a paragraph in the Discussion Section regarding cicatrization, as to why scar formation after healing in the oral cavity is less based on our study. The following paragraph is added.

1.3) This is a great study, with interesting results. Image 06 does not represent this study, but a grade school paperwork. The authors should present a professional Graphic for Image 06.

Our response: Thank you for your constructive feedback. We have revised Figure 6 to ensure a professional and polished presentation.

REVIEWER #2:

The manuscript is focused in evaluating epithelial regeneration, connective tissue proliferation and bone formation over the first week following tooth extraction in mice. The authors claim this is unprecedented data, however, the cascade of events occurring during socket healing is well documented. Maybe, their design (consecutive euthanasia over the week) might have added information, but the authors failed to make clear which gap was filled by their data. Besides this limitation that should be addressed, some other major changes need to be done prior to re-consideration for publication, once the authors succeed in

2.1) Use the background to state the singularity of osteomucosal healing and exactly which information is been added by your research. “Early events” is vague.

Our response: Thank you for your comments. We added additional background information, and we emphasized the importance of this study by stating that this study examined spatial and temporal changes in soft and hard tissues. We have revised the abstract by adding the following statement in the beginning.

“Healing in the oral cavity after dentoalveolar trauma such as tooth extraction is a unique process that involves osteomucosal healing - healing of soft and hard tissues at the same time - through hemostasis/coagulation, inflammation, proliferation, and modeling/remodeling stages. Although healing process of soft or hard tissues is well-documented previously independent to each other, the progression of simultaneous healing processes of both during the early time points remain unclear. In this study, we investigated spatial and temporal patterns of epithelial closure, collagen deposition, osteoclast formation, and woven bone development during the early stages of osteomucosal healing.”

2.2) The rationale is currently based in distinct healing across tissues. Instead, the introduction should focus on the unknown events in osteomucosal healing and plausible crosstalk between the events occurring within the socket.

Our response: Thank you for this insightful suggestion. Based on the Reviewer’s suggestion, we have revised the Introduction as following.

“Previous studies on osteomucosal healing patterns have been carried out in both humans and animals, mainly focusing on the weeks and months following an injury [6-8]. Although these studies provided valuable insights into tissue remodeling and bone regeneration, critical early events including spatial and temporal coordination of soft and hard tissue healing within the first week after extraction have been underexplored. In this study, we aim to address this knowledge gap by examining the early stages of the osteomucosal healing process within the first week following tooth extraction in mice. Specifically, we examine the kinetics of epithelial wound closure, connective tissue formation, and new bone formation, as well as monitor the formation of osteoclasts and the expression of OPG and RANKL at the tooth extraction sites.”

2.3). Did you perform sample size calculation? Based on most literature of socket healing in mice, N=3 seems not sufficient to assure relevance. You mention that both right and left first molars were extracted, which is different from the abstract. Which one is correct?

Our response: Thank you for asking this clarification. In our study, N=3 refers to the number of mice per group, with both the maxillary right and left first molars extracted from each mouse, resulting in 6 total extraction sockets per group at each time point. To ensure clarity, we have revised the Materials and Methods section to explicitly state that each group consisted of 6 sockets (2 sockets per mouse × 3 mice) as the following.

“Three mice were sacrificed every day for 7 days, resulting in a total of 6 extraction sockets per group per time point (2 sockets per mouse × 3 mice per time point).”

2.4) Describe how you defined the region of interest and the method of analysis for the microCT. Mason’s Trichrome was only mentioned in the results and the information is lacking in the M&M. The method of analysis for all outcome parameters was neglected or poorly described. An extensive revision on that is necessary.

Our response: Thank you for pointing out the need for more detailed descriptions of our methodologies. We have revised the Materials and Methods section as the following.

“The Region of Interest (ROI) was defined as the extraction socket, extending from the most coronal aspect of the alveolar bone crest to the apical portion of the socket. The ROI was segmented based on grayscale thresholding to differentiate between mineralized and non-mineralized tissues.”

2.5) The clinical analysis should focus on wound closure only, instead of describing tissues. Epithelium can be appreciated in the histological slides. The word “matured” should be changed to proper histologic terminology. The figures are shown in poor quality, hindering a proper appreciation of their representativeness. Additionally, they are presented in low magnification, which compromises a proper visualization of the tissue events described in the text. The IHC is also shown in low magnification. I understand that the labeling is not confined to the cytosolic compartment since these proteins exert their function when secreted in the extracellular matrix. However, higher magnification images are necessary to identify what is antigen-antibody labeling and what is background labeling.

Our response: Thank you for your detailed feedback. We have made significant revisions to address these points:

 Focus on Wound Closure: In the revised Results section, we have focused the analysis on wound closure dynamics and removed unnecessary descriptions of the surrounding tissues.

 Terminology Update: The term "matured" has been replaced with proper histological terminology. Specifically, we now use terms like "fully developed connective tissue" to align with standard histological language.

 Improved Figure Quality: The current magnification was deliberately chosen to capture the entire socket area and provide an overview of the morphological changes occurring during the healing process. We recognize that this broader view may lack the detailed resolution needed for certain features, so we have added additional higher-magnification images of key areas to provide greater clarity.

2.6) Overall, the discussion is missing comprehensive understanding of the data shown by the authors. Also, the authors could explore the sequential (and overlapping) events trying to establish a spatial relevance of uneventful osteomucosal healing.

Our response: Thank you for your valuable feedback. In the Discussion Section, we have revised accordingly so that we discuss about the spatial and temporal events occurred during osteomucosal healing. In particular, we discussed about epithelial closure first (2nd paragraph), then collagen formation (2nd paragraph) followed by bone formation (3rd paragraph). After that, we focused on discussion about osteoclasts (4th and 5th paragraphs), explanation of which is foundational to our further discussion about 3 clinical implications – dry socket, MRONJ, and scarless formation.

2.7) A proper conclusion based on the results should be drawn.

Our response: Thank you for your constructive feedback. We have included the Conclusion paragraph at the end of the Discussion Section.

REVIWER #3:

The proposed paper details the first week of alveolar repair in daily analyses. A very interesting study that will certainly support studies regarding interferences in the natural course. The objectives are clear and the methodology is consolidated in the literature in relation to alveolar repair.

It precisely details the role of osteoclasts in the early stages after tooth extraction, which is well explored in the discussion as possible indicators for studies of the mechanisms of MRONJ.

Based on the analysis of the study, I suggest some small changes:

3.1) There is a tendency in the results section to repeat some points from the methods, for example:

- “To investigate the kinetics of collagen formation and deposition following tooth extraction, we stained the sections with Trichrome, a maker of collagens in general…”

I suggest that these points be part of the method only and not repeated in the results section.

Our response: Thank you for your feedback. We have revised the Results section to reflect the Reviewer’s suggestion. Here is one example.

Previous version:

“To investigate the kinetics of collagen formation and deposition following tooth extraction, we stained the sections with Trichrome, a maker of collagens in general, and found that collagen deposition occurs as early as day 2 (Fig. 2A). Picro-Sirius Red staining revealed that collagen type I (yellow/organge/red) and III (green) followed a similar pattern; they started occurring as early as day 2 post-extraction and exponentially increased by day 7 (Fig. 2B and 2C).”

Revised version:

“Collagen deposition was observed as early as day 2 post-extraction (Fig. 2A, top row). Further analysis revealed that both collagen type I (yellow/organge/red) and III (green) followed a similar temporal pattern, appearing on day 2 and increasing exponentially by day 7 (Fig 2A, middle and bottom rows; Fig. 2B and 2C). Statistical analyses across time points indicated a significant increase in collagen deposition between day 6 and day 7 (p < 0.05), indicating rapid extracellular matrix production during this period.”

2. Regarding Figure 6: although it seems obvious that the markings on the horizontal axis refer to the 7 days studied, I suggest that the numbers of the days to which the markings refer be written. In the same way, markings be added to the vertical axis.

Our response: Thank you for the comments. We modified the graph as your suggestion.

3. In the caption of Figure 1, I suggest removing "N = 6 tooth extraction sites with 3 mice per group.", This refers to the method and is already very clear in the text.

Our response: Thank you. We deleted this statement. Also, please refer to Reviwer #2 Q3 for further clarification.

Reviewer #4:

Kim et al. presented an original study investigating early cellular and molecular events in osteomucosal healing in a mouse model. This study provides a new understanding of collagen deposition, osteoclast activity, and remodeling. The results have clear clinical implications for MRONJ and dry sockets using advanced imaging and staining techniques. Overall, the manuscript provides valuable contributions to the understanding of osteomucosal healing. Its clarity and impact will be further enhanced by addressing the following suggestions.

4.1) Abstract: While the abstract is informative, it lacks brief details on the materials and methods used. Including this information would improve clarity and help readers understand how the results were obtained.

Our response: Thank you for this suggestion. We have revised the abstract to include brief details of the materials and methods used in our study.

4.2) Although the figures and results are robust, the discussion could be more explicit in linking the results to potential future studies on clinical implications, such as MRONJ or dry socket.

Our response: We have revised the discussion to include explicit connections between our findings and potential clinical implications as below.

“These observations have several important clinical implications. First, tooth extraction often leads to alveolar osteitis (AO) also known as “dry socket.” AO commonly occurs when the blood clots become partially or totally dislodged from the socket, and healing of the dry socket may last several weeks [27]. Based on our study, it is possible that the lodged blood clots may remove an opportunity to induce de novo bone formation in an osteoclast-independent manner, leaving the bone formation mode only from the existing bone surfaces that is osteoclast-dependent, a process of which can be slow and further delays the proper osteomucosal wound healing process. Second, in patients taking anti-resorptive agents such as bisphosphonates or denosumab, the osteoclast-osteoblast coupling mechanism is compromised due to inhibition of osteoclast formation and/or functions [28]. As such, bone formation in the socket is primarily dependent on de novo bone formation in the blood clots. In these patients, if the blood clots become dislodged after tooth extraction, or if bleeding does not get induced, it may lead to medication-related osteonecrosis of the jaw (MRONJ). Lastly, wound healing in the oral cavity is known for expedited healing with minimal cicatrization, and this scar formation is primarily mediated by dysregulated production of collagens by fibroblasts. Recent studies suggest that microanatomic differences of fibroblasts lead to scar formation; fibroblasts residing in the deep layers are more prone to cause scar formation when compared to the those in the superficial layers [2]. As such, it is possible to speculate that deeper fibroblasts may contribute more to bone formation, rather than scar formation, that leads scarless healing unique to the oral cavity. Further studies await confirmations of these interpretations.”

4.3) The narrative would be streamlined, and readability would be improved by reducing redundancy in the introduction and discussion.

Our response: We have streamlined and extensively revised the introduction and the discussion by removing redundant statements and reorganizing the text for better readability.

4.4) This study did not use additional molecular methods, such as qRT-PCR for gene expression analysis, that could have deepened the findings.

Our response: Thank you for this insightful comment. We agree to the reviewer, and we have changed the title to limit to cellular events only. While our current study focused on histological and immunohistochemical characterization of early osteomucosal healing, we recognize the importance of molecular methods to further elucidate the underlying mechanisms. To this end, we have already established a follow-up study using single-cell RNA sequencing (scRNA-seq) to investigate gene expression at the cellular level at 1 week after extraction (Revision Fig. 1). This approach will allow us to identify specific cell populations, and their gene expression profiles, providing a more detailed understanding of the molecular pathways driving osteomucosal healing. We add this figure for the reviewer purposes only.

---

## [Decision Letter · Decision Letter 1]

23 Apr 2025

PONE-D-24-55861R1Early cellular events of osteomucosal healing in the tooth extraction socketPLOS ONE

Dear Dr. Kim,

Thank you for submitting your manuscript to PLOS ONE. After careful consideration, we feel that it has merit but does not fully meet PLOS ONE’s publication criteria as it currently stands. Therefore, we invite you to submit a revised version of the manuscript that addresses the points raised during the review process.

We look forward to receiving your revised manuscript.

Kind regards,

Henrique Hadad, DDS, Ph.D.

Academic Editor

PLOS ONE

Additional Editor Comments:

Dear Authors,

Thank you for your submission. I believe your manuscript presents an interesting contribution to the field. However, I remain concerned about the Methodology section and certain aspects of the Results.

Please find attached the reviewers' comments for your consideration. Both reviewers expressed concern that the manuscript does not adequately address the stated aim of the study, noting that it "does not provide sufficient depth to fully elucidate the regulatory mechanisms underlying socket healing."

I strongly encourage you to address these points thoroughly and revise your manuscript accordingly before resubmitting it for further evaluation.

Reviewers' comments:

Reviewer's Responses to Questions

**Comments to the Author**

1. If the authors have adequately addressed your comments raised in a previous round of review and you feel that this manuscript is now acceptable for publication, you may indicate that here to bypass the “Comments to the Author” section, enter your conflict of interest statement in the “Confidential to Editor” section, and submit your "Accept" recommendation.

Reviewer #2: (No Response)

Reviewer #6: (No Response)

Reviewer #7: (No Response)

2. Is the manuscript technically sound, and do the data support the conclusions?

Reviewer #2: Yes

Reviewer #6: Partly

Reviewer #7: Partly

3. Has the statistical analysis been performed appropriately and rigorously? 

Reviewer #2: Yes

Reviewer #6: Yes

Reviewer #7: Yes

4. Have the authors made all data underlying the findings in their manuscript fully available?

Reviewer #2: Yes

Reviewer #6: Yes

Reviewer #7: Yes

5. Is the manuscript presented in an intelligible fashion and written in standard English?

Reviewer #2: Yes

Reviewer #6: Yes

Reviewer #7: (No Response)

6. Review Comments to the Author

Reviewer #2: The authors have implemented significant revisions to the manuscript, notably enhancing its quality. Among the previous comments provided by this reviewer, the authors have neglected to address one particular issue related to the methods of analysis for the histological and histochemical data. It is recommended that the authors include the detailed methods of analysis for each of the outcome measures evaluated, not limited to microCT.

Although the manuscript is written in formal and intelligible English, a comprehensive revision is necessary to resolve the grammatical inconsistencies that have emerged during the review process.

Reviewer #6: The authors set out to investigate the spatial and temporal changes in epithelial, connective and bone tissues, in addition to the presence of osteoclasts during the early stages of osteomucosal healing.

I have seen that this article has already been reviewed by extremely meticulous reviewers, which has certainly made it easier for the current reviewer to make his considerations.

First of all, I would like to point out that the main objective, which was to investigate spatial and temporal changes in epithelial, connective and bone tissues, was not fully achieved. Epithelial tissue was not included in either the methodology or the results. Information on epithelial proliferation and maturation of epithelial layers with keratin formation was not analysed. Immunohistochemistry for epithelial markers such as CKPool and EGFR would have enriched the study. In addition, temporal changes were only made between connective tissue and new bone formation.

Materials and Methods

The authors said: “Atraumatic extraction of the maxillary first molars (M1) was carried out”. However, rodent maxillary first molars have a variable number of roots and are very thin. This reviewer is curious to know whether root fractures were found. This is important information to clarify for the reader.

Immunohistochemical staining

It is important to describe the immunohistochemical evaluation parameters, including the cellular localisation of expression of the two antibodies evaluated. RANK-L is expressed at the cell membrane and in the cytoplasm. Do the authors take this location into account when scoring the samples? What about the anti-osteoprotegerin antibody? It is usually secreted.

Results

Figure 1(D) Day 3 - I would suggest revising the description of the histological findings. It is possible to see a proliferation of spindle cells at a higher magnification, and this data is important for the healing process.

Figure 1(D) Day 4 - I suggest replacement of the microphotograph from this day, as the histological aspects are worse in the healing process than on day 3. Do the basophilic areas suggest necrosis or the accumulation of polymorphonuclear neutrophils cells?

Figure 1(D) Day 5 - This photomicrograph shows the proliferative pattern of spindle cells. However, there are a lot of wood shavings remaining in the upper field, which could hinder the healing process. Do the authors have another image with no wood shavings to replace it?

Figure 1(D) Days 6 and 7 were good.

With regard to Figure 2, this reviewer did not see any temporal changes in the alveoli, with Mason's trichrome staining in terms of new bone formation, except on days 6 and 7.

On the other hand, PS Red staining, with and without polarised light, did not contribute to the analysis and actually exacerbated the visualisation of wood shavings within the alveoli on days 4 and 5.

The durability of this set of micrographs should be re-evaluated.

The authors describe this in the Results section: “Oral mucosal tissue closure following tooth extraction in mice”, “By day 1, the blood clot was replaced by a granulation tissue that seemingly consisted of three distinct layers: the residual coagulum layer in the apical portion, the loose provisional matrix layer in the middle of the socket, and the neo-epithelial layers in the coronal portion.”

In pathology, granulation tissue is a vascular connective tissue that forms in wounds during the healing process. It's a hallmark of wound healing and is characterised histologically by the presence and proliferation of fibroblasts, macrophages, keratinocytes, endothelial cells, new thin-walled capillaries and inflammatory cell infiltration of the extracellular matrix.

The presence of granulation tissue in a wound means that the inflammatory phase has passed and the proliferative phase has begun. It's important to remember that under normal conditions, wound healing and tissue repair occur in 4 stages: haemostasis, inflammatory stage, proliferative stage (when granulation tissue formation occurs) and remodelling stage.

Therefore, it is not possible to say that the blood clot has been replaced by granulation tissue on day 1.

Finally, this reviewer would like to know why the teeth in the tomographic and histological images have their roots in the lower part of the image and the crowns in the upper part, simulating teeth from the lower arch. Have the upper/maxillary first molars not been extracted? If so, the roots should be in the upper part of the images.

Please correct these words:

Such difference is largely owing to multiple layers of in-beween (in-between) anatomical structures such as muscle, fat, or fascia that separates soft and hard tissues.

…..analysis revealed that both collagen type I (yellow/organge (orange)/red) and III (green) followed a similar……

Reviewer #7: This revised manuscript investigates the early healing process in tooth extraction sockets. As the authors note, morphological analyses provide valuable insights into post-extraction healing. The selected methodologies are generally appropriate; however, the resolution and depth of the observations are limited, and as such, they fall short of offering substantial mechanistic insight into the healing process. Detailed comments are provided below.

1. Oral and Mucosal Tissue Closure Following Tooth Extraction in Mice (Figure 1)

The authors describe bone resorption mediated by osteoclasts; however, based on the current image resolution, osteoclasts cannot be clearly identified.

Figure 1D: The image size and resolution are insufficient to support the conclusions drawn in the results. Since these images are foundational to the study’s claims, they should be presented in higher resolution and at larger sizes to improve clarity and impact.

2. Collagen Deposition in the Tooth Extraction Sockets (Figure 2)

The rationale for this analysis, particularly the distinction between Type I and Type III collagen, needs to be explicitly stated. Additionally, a detailed explanation of the spatial distribution of these collagen types within the healing socket should be provided.

While Picrosirius Red staining under polarized light is commonly used to assess collagen types, it is not definitive. Therefore, confirmation via immunohistochemistry is necessary.

The image quality, especially under polarized microscopy, is not sufficient to clearly distinguish collagen types or their distribution. Larger, higher-resolution images are required to support the data.

3. Woven Bone Formation in the Tooth Extraction Sockets (Figure 3)

The purpose of this analysis should be clearly stated. In the discussion, the authors note the continuity between woven and pre-existing bone, and the woven bone is formed independently within the healing connective tissue. It is an important observation, and thus should be explicitly addressed here. Micro-CT 3D reconstruction could clarify whether distinct bone formation, such as appositional growth on existing bone surfaces versus de novo intramembranous ossification, is occurring. Additionally, further exploring these distinct processes associated with molecular regulation would greatly enhance the depth of the study.

In Panel A, the distobuccal (DB) socket appears to exhibit earlier woven bone formation compared to the mesial (M) socket. The authors should acknowledge this and analyze whether healing differs significantly between the two sites.

4. Osteoclast Formation in the Tooth Extraction Sockets (Figure 4)

To better differentiate between osteoclasts associated with newly formed versus pre-existing bone, magnified insets of these respective areas should be provided. This would allow clearer visualization of osteoclast localization relative to bone context.

5. Expression of RANKL and OPG in the Tooth Extraction Sockets (Figure 5)

Since RANKL is produced not only by osteoblasts but also by lymphocytes and stromal cells, the authors should carefully delineate the association of RANKL-positive staining with specific cell types and tissue structures.

To confirm the relationship between RANKL expression and osteoclast activity, double staining for RANKL and TRAP is recommended.

6. Overall Comments

The current dataset does not provide sufficient depth to fully elucidate the regulatory mechanisms underlying socket healing. Given the breadth of existing literature on bone wound healing, the authors should contextualize their findings with key molecular events known to occur during skeletal repair.

Moreover, tooth extraction results in disruption and torsion of collagen fibers from the periodontal ligament (PDL) that insert into the alveolar bone, likely causing structural changes to the bone surface. The mechanical load during tooth extraction may also affect osteocytes adjacent to the bone surface. Beyond the observed difference in the osteoclast distribution, the authors need to demonstrate the possible causes that could influence osteoclast recruitment and socket bone remodeling during the healing process. Investigating these structural and cellular dynamics could offer important insights into the interplay between immune responses, stromal cells, and osteoclastogenesis in the healing socket.

7. PLOS authors have the option to publish the peer review history of their article (what does this mean?). If published, this will include your full peer review and any attached files.

Reviewer #2: No

Reviewer #6: No

Reviewer #7: No

---

## [Author Response · Author response to Decision Letter 2]

28 May 2025

May 27, 2025

Dr. Henrique Hadad

Academic Editor,

PLoS ONE

We received your correspondence of April 23rd, 2025, and the comments of the referees concerning our manuscript “Early cellular events of osteomucosal healing in the tooth extraction socket" by Kim et al. (PONE-D-24-55861R1) submitted to PLoS ONE. We deeply appreciate the referees’ comments. Based on their comments, we have modified the manuscript to correctly address all their concerns. The revisions included in the manuscript are described in our point-by-point responses to the referees’ comments detailed below.

Review Comments to the Author

Reviewer #2:

The authors have implemented significant revisions to the manuscript, notably enhancing its quality. Among the previous comments provided by this reviewer, the authors have neglected to address one particular issue related to the methods of analysis for the histological and histochemical data. It is recommended that the authors include the detailed methods of analysis for each of the outcome measures evaluated, not limited to microCT.

Although the manuscript is written in formal and intelligible English, a comprehensive revision is necessary to resolve the grammatical inconsistencies that have emerged during the review process.

Our response: Thank you for your suggestion. In response to your suggestion, we have thoroughly revised and expanded the "Materials and Methods" section to include comprehensive and detailed descriptions of the methodologies used for analyzing our histological and histochemical outcomes, including Trichrome and Picrosirius Red staining procedures. Additionally, we have carefully performed a comprehensive grammatical review of the manuscript to address and rectify inconsistencies that arose during the previous revision processes.

Reviewer #6:

The authors set out to investigate the spatial and temporal changes in epithelial, connective and bone tissues, in addition to the presence of osteoclasts during the early stages of osteomucosal healing.

I have seen that this article has already been reviewed by extremely meticulous reviewers, which has certainly made it easier for the current reviewer to make his considerations.

First of all, I would like to point out that the main objective, which was to investigate spatial and temporal changes in epithelial, connective and bone tissues, was not fully achieved. Epithelial tissue was not included in either the methodology or the results. Information on epithelial proliferation and maturation of epithelial layers with keratin formation was not analysed. Immunohistochemistry for epithelial markers such as CKPool and EGFR would have enriched the study. In addition, temporal changes were only made between connective tissue and new bone formation.

Our response: Thank you for raising this important point. While epithelial proliferation and maturation of epithelial layers are important parts of soft tissue healing, we mainly focused on epithelial migration (as determined by wound closure) because it is one of the main clinical criteria of determining whether the tooth extraction sockets are healing or not. Indeed, 1-2 weeks of post-op appointment in a dental office is to examine whether there is any bone exposure remaining. In the normal setting, epithelial proliferation and maturation (e.g., keratinization) follow when epithelial migration occurs successfully and closes the wounded site. Therefore, our focus in investigating the cellular changes in osteomucosal healing is to determine the wound closure as shown in the Fig. 1. Currently, we are working on changes in osteomucosal healing at the molecular level using single cell RNA sequencing (scRNA-seq), preliminary results of which were briefly shared in our previous revised version. We do agree that these molecular analyses will further enrich the study, but we also feel that adding these analyses will not change the conclusion of our study and are beyond the scope of this manuscript. This insight offers meaningful guidance for our future studies.

Materials and Methods

The authors said: “Atraumatic extraction of the maxillary first molars (M1) was carried out”. However, rodent maxillary first molars have a variable number of roots and are very thin. This reviewer is curious to know whether root fractures were found. This is important information to clarify for the reader.

Our response: Clinically, atraumatic extraction in human can be performed very predictably and reproducibly when elevators/luxators are effectively used to separate periodontal ligaments (PDL) before using forceps. In mice, we use only explorer, but it is being used as both an elevator/luxator and a forcep. This can be done by allowing enough time to elevate the tooth by engaging at the buccal furcation area of the first molar like an elevator/luxator. When you start “feel” loosening of the tooth from the socket, the tip of the explorer can be rotated to extract the tooth like a forcep. Our laboratory has extensive experience performing mouse tooth extractions, having routinely carried out this procedure for over 10 years. Thus, we are comfortable in the reproducibility and reliability of our extraction methods without root fractures. Although it is extremely rare for us by now, we do seldomly experience root fractures mostly on the M root due to angulation issues, especially when the bone becomes very strong by anti-resorptive agents such as bisphosphonates or anti-RANKL neutralizing antibody. In this study, however, we did not experience root fracture. We have explicitly clarified this point in the revised Materials and Methods section of the manuscript to ensure complete transparency and to avoid any potential confusion for readers as the following (high-lighted below).

“μCT Scan

Areas of interest on the maxillae were subjected to μCT scanning (SkyScan1275, Bruker, Kontich, Belgium) at 60 kVp and 166 μA using a voxel size of 10 μm3 and a 0.5 mm Aluminum filter with an integration time of 200 ms using a cylindrical tube (FOV/Diameter: 20.48 mm). The Region of Interest (ROI) was defined as the extraction socket, extending from the most coronal aspect of the alveolar bone crest to the apical portion of the socket. Fractured root tips were observed from the μCT images and confirmed no remaining root tips in all analyzed sockets. The ROI was segmented based on grayscale thresholding to differentiate between mineralized and non-mineralized tissues. Two-dimensional slices from each femur were combined using the SkyScan NRecon software to form a three-dimensional reconstruction. The image analysis was performed using the DataViewer and CTAn software.”

Immunohistochemical staining

It is important to describe the immunohistochemical evaluation parameters, including the cellular localisation of expression of the two antibodies evaluated. RANK-L is expressed at the cell membrane and in the cytoplasm. Do the authors take this location into account when scoring the samples? What about the anti-osteoprotegerin antibody? It is usually secreted.

Our response: Thank you for this thoughtful comment. In our immunohistochemical evaluation, we carefully considered the localization patterns of both RANKL and OPG when interpreting the staining results:

For RANKL, positive staining was observed in both the cytoplasm and the cell membrane of stromal and mononuclear cells adjacent to the bone surfaces. Both patterns of subcellular localization were taken into account in our interpretation, as described in the revised Results section. RANKL expression was particularly prominent during the early healing phase (Days 0–4) and became more restricted and less intense by Days 6–7, consistent with its role in initiating osteoclastogenesis during early socket remodeling.

For OPG, consistent with its biological function as a secreted soluble factor, staining appeared primarily as the diffuse extracellular matrix-associated signal, especially surrounding newly formed bone. Rather than scoring individual cell positivity, we interpreted OPG expression based on its distribution pattern and localization within the tissue microenvironment.

Our intent with these IHC experiments was to illustrate the spatial relationship between these two regulators and osteoclast activity over time.

Results

Figure 1(D) Day 3 - I would suggest revising the description of the histological findings. It is possible to see a proliferation of spindle cells at a higher magnification, and this data is important for the healing process.

Figure 1(D) Day 4 - I suggest replacement of the microphotograph from this day, as the histological aspects are worse in the healing process than on day 3. Do the basophilic areas suggest necrosis or the accumulation of polymorphonuclear neutrophils cells?

Figure 1(D) Day 5 - This photomicrograph shows the proliferative pattern of spindle cells. However, there are a lot of wood shavings remaining in the upper field, which could hinder the healing process. Do the authors have another image with no wood shavings to replace it? Figure 1(D) Days 6 and 7 were good.

Our response: Thank you for your detailed feedback on the histological data. To address your suggestions comprehensively, we have retaken and replaced the original images with new micrographs captured under higher resolution and magnification. Specifically:

Day 3: The revised images now clearly illustrate the proliferation of spindle cells under higher magnification. This cellular detail is now explicitly highlighted and described in the revised results section, providing enhanced clarity and stronger support for our conclusions.

Day 4: We have provided a new microphotograph that better represents the healing stage observed on Day 4.

Day 5: We replaced the original photomicrograph with a new image that is free from wood shaving artifacts, allowing clearer visualization of the proliferative spindle cell pattern.

With regard to Figure 2, this reviewer did not see any temporal changes in the alveoli, with Mason's trichrome staining in terms of new bone formation, except on days 6 and 7.

On the other hand, PS Red staining, with and without polarised light, did not contribute to the analysis and actually exacerbated the visualisation of wood shavings within the alveoli on days 4 and 5.

The durability of this set of micrographs should be re-evaluated.

Our response: Thank you for your detailed observation. In response to your comment, we have retaken the Trichrome staining and Picrosirius Red-stained images at higher resolution and magnification to improve visual clarity and minimize interference from wood shavings. The updated images now provide a clearer view of tissue composition and more clearly distinguish temporal changes in the healing sockets.

Additionally, we re-quantified collagen type I and type III deposition using high-magnification images to improve accuracy and enhance the interpretability of the results. We believe these updates significantly improve the quality and reliability of Figure 2 and more effectively illustrate the progression of bone and collagen matrix formation during the healing process.

The authors describe this in the Results section: “Oral mucosal tissue closure following tooth extraction in mice”, “By day 1, the blood clot was replaced by a granulation tissue that seemingly consisted of three distinct layers: the residual coagulum layer in the apical portion, the loose provisional matrix layer in the middle of the socket, and the neo-epithelial layers in the coronal portion.”

In pathology, granulation tissue is a vascular connective tissue that forms in wounds during the healing process. It's a hallmark of wound healing and is characterised histologically by the presence and proliferation of fibroblasts, macrophages, keratinocytes, endothelial cells, new thin-walled capillaries and inflammatory cell infiltration of the extracellular matrix.

The presence of granulation tissue in a wound means that the inflammatory phase has passed and the proliferative phase has begun. It's important to remember that under normal conditions, wound healing and tissue repair occur in 4 stages: haemostasis, inflammatory stage, proliferative stage (when granulation tissue formation occurs) and remodelling stage.

Therefore, it is not possible to say that the blood clot has been replaced by granulation tissue on day 1.

Our response: Thank you for your important and insightful comment. We agree that the classical definition of granulation tissue refers to a vascularized connective tissue formed during the proliferative phase of wound healing, following the inflammatory phase. Based on your observation, we recognize that it was premature to refer to the Day 1 wound content as granulation tissue. Accordingly, we have revised the Results section to more accurately describe the Day 1 findings as follows:

"By Day 1, the extraction socket was filled with a residual blood clot, transitioning toward the formation of a provisional matrix layer consisting of early inflammatory cell infiltration and initial extracellular matrix deposition."

We appreciate the reviewer’s clarification and have modified the language throughout the manuscript to ensure alignment with standard pathological definitions of wound healing stages.

Finally, this reviewer would like to know why the teeth in the tomographic and histological images have their roots in the lower part of the image and the crowns in the upper part, simulating teeth from the lower arch. Have the upper/maxillary first molars not been extracted? If so, the roots should be in the upper part of the images.

Our response: Thank you for this helpful observation. We confirm that all images in the study represent maxillary first molars. To improve consistency and ease of interpretation, we chose to orient the tomographic and histological images with the crown at the top and the roots at the bottom across all time points. In the field of medicine where skin wounds and healing are being described, basal cell layers are typically placed in the bottom while keratinized layers are placed on the top regardless of the human anatomical parts (e.g., sole foot). Similarly, in dentistry, this orientation of epithelial tissues and tooth crown being on the top while bone tissues and roots being in the bottom mirrors the layout commonly used in histological sections and allows for easier side-by-side comparisons between modalities.

To prevent confusion, we have revised the figure legends and relevant methods text to explicitly state that these images are from the maxillary first molars and that the orientation was standardized for visual clarity.

Please correct these words:

Such difference is largely owing to multiple layers of in-beween (in-between) anatomical structures such as muscle, fat, or fascia that separates soft and hard tissues.

…..analysis revealed that both collagen type I (yellow/organge (orange)/red) and III (green) followed a similar……

Our response: Thank you for carefully reviewing our manuscript and highlighting these typographical errors. We have now corrected these mistakes. Additionally, we have carefully reviewed the entire manuscript again to ensure no further typographical errors remain. 

Reviewer #7:

This revised manuscript investigates the early healing process in tooth extraction sockets. As the authors note, morphological analyses provide valuable insights into post-extraction healing. The selected methodologies are generally appropriate; however, the resolution and depth of the observations are limited, and as such, they fall short of offering substantial mechanistic insight into the healing process. Detailed comments are provided below.

1. Oral and Mucosal Tissue Closure Following Tooth Extraction in Mice (Figure 1)

The authors describe bone resorption mediated by osteoclasts; however, based on

---

## [Decision Letter · Decision Letter 2]

2 Jul 2025

Early cellular events of osteomucosal healing in the tooth extraction socket

PONE-D-24-55861R2

Dear Dr. Kim,

We’re pleased to inform you that your manuscript has been judged scientifically suitable for publication and will be formally accepted for publication once it meets all outstanding technical requirements.

Kind regards,

Henrique Hadad, DDS, Ph.D.

Academic Editor

PLOS ONE

Additional Editor Comments (optional):

Reviewers' comments:

Reviewer's Responses to Questions

**Comments to the Author**

1. If the authors have adequately addressed your comments raised in a previous round of review and you feel that this manuscript is now acceptable for publication, you may indicate that here to bypass the “Comments to the Author” section, enter your conflict of interest statement in the “Confidential to Editor” section, and submit your "Accept" recommendation.

Reviewer #6: All comments have been addressed

Reviewer #7: (No Response)

2. Is the manuscript technically sound, and do the data support the conclusions?

Reviewer #6: Yes

Reviewer #7: Yes

3. Has the statistical analysis been performed appropriately and rigorously? 

Reviewer #6: Yes

Reviewer #7: Yes

4. Have the authors made all data underlying the findings in their manuscript fully available?

Reviewer #6: Yes

Reviewer #7: Yes

5. Is the manuscript presented in an intelligible fashion and written in standard English?

Reviewer #6: Yes

Reviewer #7: Yes

6. Review Comments to the Author

Reviewer #6: After analyzing the updated version of the article, I found that it had been enhanced with more detailed and relevant information. Many suggestions made by different reviewers were considered and incorporated, resulting in a more substantial article. The current dataset may help clarify the mechanisms of dental alveolar healing, despite the lack of some analyses.

I believe that the manuscript meets the standards for publication of this prestigious scientific journal, but I would like to leave some comments for the authors.

I realize that I was unclear about the spatial and temporal aspects of changes in epithelial, connective, and bone tissues. I expected the authors to describe only the histological aspects of epithelial tissue during phases when collagen and bone tissue were deposited. This would provide readers with a spatial and temporal understanding of the relationship between these tissues in each analyzed phase. Besides reviewing the hematoxylin and eosin staining, no other methodology would be necessary.

Epithelial tissue is only mentioned on day four, when the authors wrote, "At the same time, epithelial layers began migrating toward the wound..." What about the other days? I believe it would be very interesting if the opportunity arises and there is enough time.

Reviewer #7: This revision addresses most of the reviewer’s concerns; however, the rationale and significance behind the respective evaluation of Type I and Type III collagens still require further clarification. In the current results and discussion, these two collagen types are collectively referred to simply as “collagens,” despite their known distinct roles in wound healing. Given that the authors specifically highlight Type I and Type III collagens using polarized light microscopy, it is important to explain the reasoning behind this approach and to interpret the findings accordingly. The discussion should elaborate on their respective spatial distributions, especially if distinct localization patterns were observed. A more detailed comparison of the similarities and differences between Type I and Type III collagens in the context of the healing process would enhance the depth and impact of this study.

7. PLOS authors have the option to publish the peer review history of their article (what does this mean?). If published, this will include your full peer review and any attached files.

Reviewer #6: No

Reviewer #7: No

---

## [Editor Report · Acceptance letter]

PONE-D-24-55861R2

PLOS ONE

Dear Dr. Kim,

I'm pleased to inform you that your manuscript has been deemed suitable for publication in PLOS ONE. Congratulations! Your manuscript is now being handed over to our production team.

Kind regards,

on behalf of

Dr. Henrique Hadad

Academic Editor

PLOS ONE